# The Development of a Remote Edge-Lit Backlight Structure with Blue Laser Diodes

**Bing-Mau Chen [1], Shang-Ping Ying [1,\*] , Truong An Pham [1], Shiuan-Yu Tseng [2] and Yu-Kang Chang [3,4,5]**

1   Department of Semiconductor and Electro-Optical Technology, Minghsin University of Science & Technology, 1, Xinxing Road, Xinfeng, Hsin-Chu 30401, Taiwan; bmchen@must.edu.tw (B.-M.C.); b09130018@std.must.edu.tw (T.A.P.)
2   Department of Senior Services Industry Management, Minghsin University of Science & Technology, 1, Xinxing Road, Xinfeng, Hsin-Chu 30401, Taiwan; sytseng@must.edu.tw
3   Department of Medical Research, Tungs' Taichung MetroHarbor Hospital, Taichung City 435, Taiwan; t12193@ms3.sltung.com.tw
4   Department of Post-Baccalaureate Medicine, College of Medicine, National Chung Hsing University, Taichung 402202, Taiwan
5   Department of Nursing, Jenteh Junior College of Medicine, Nursing and Management, Miaoli 356006, Taiwan
\*   Correspondence: sbying@must.edu.tw

**Abstract:** In this study, we introduce a novel design of a remote edge-lit backlight structure featuring blue laser diodes (LDs). These LDs were integrated into a remote yellow phosphor layer on a light guide plate (LGP). Blue light emitted by the LDs passes through the LGP and spreads to the remote phosphor layer, generating white light output. Owing to the incorporation of a scattering layer between sequential LGPs, the remote edge-lit backlight structure facilitates the expansion of the output surface of the LGP by combining multiple individual LGPs. Two- and three-LGP remote edge-lit backlight structures demonstrated acceptable white illuminance uniformity. The proposed architecture serves as a viable solution for achieving uniform illumination in planar lighting systems using blue LDs; thus, this structure is particularly suitable for linear lighting or slender backlighting instead of display stand applications.

**Keywords:** laser diodes (LDs); edge-lit backlight; phosphor layer; planar lighting system

## 1. Introduction

Flat panel displays (FPDs) are highly versatile and have become indispensable in various display applications, including entertainment, mobile devices, and computer monitors. The increasing demand for information and interactive FPDs has resulted in a growing need for electronic devices, such as laptops, mobile phones, and tablet televisions equipped with FPDs. Currently, there are two major FPD technologies: organic light-emitting diode (OLED) display and liquid crystal display (LCD) [1–4]. OLED displays offer advantages such as panel flexibility, high contrast ratios, and wide viewing angles [1]. However, challenges such as screen burn-in, susceptibility to humidity, rapid material aging, and high production costs have limited their widespread use in the current FPD market [5,6]. By contrast, LCDs, which are known for their affordability, long lifespan, and low power consumption, continue to dominate the current display technology market [2,7–9].

Unlike OLED displays, LCDs require a backlight, which plays a crucial role in determining the display's optical efficiency, contrast ratio, and color gamut [9,10]. Two typical LED backlight architectures are commonly used in LCDs: edge-lit and direct-lit backlights. In the edge-lit backlight technology, light sources are placed along the sides of a light guide plate (LGP). Light is emitted through the LGP and a back reflector sheet to ensure uniform light output from the top surface of the LGP. By contrast, in the direct-lit technology, light sources are arranged in a regular pattern on the backlight's backplane, forming an array that emits light directly. Nowadays, white LEDs are most commonly used for backlighting.

Direct-lit backlights offer the advantage of local dimming for specific areas of brightness on the screen, substantially enhancing dynamic contrast. However, achieving uniform light distribution with direct-lit backlights often results in a thicker backlight profile and necessitates the use of a greater number of LEDs [11]. To reduce the thickness of direct-lit backlights, extensive efforts have been made to maintain high optical performance, including specific optical designs of the light guide layer with a microstructure of concave parabel-surface microlens, reflective dots on the top surface of the LGP, and multiple three-dimensional diffuse reflection cavity arrays [12–14]. Different hybrid backlight structures that combine the elements of both edge-lit and direct-lit technologies are also used to reduce the thickness of direct-lit backlights [15–17]. An alternative approach to white light, which are also used in conventional backlights, involves a remote structure or blue-light-excited planar lighting system for direct-lit backlights that comprises a phosphor sheet, remote phosphor converter, or quantum dot converter excited by blue LEDs, as depicted in Figure 1a [18–23]. In this setup, an array of blue LEDs is used to excite the remote phosphor sheet, generating an area of white light output for direct-lit backlighting. Alternatively, in remote edge-lit backlight structures, blue LEDs emit light through the LGP, which then spreads to a remote phosphor sheet covering the LGP, providing a uniform light output, as shown in Figure 1b [20]. Although remote edge-lit backlights can be used to reduce the thickness of the backlight module, the high beam divergence of conventional LEDs makes it challenging to control optical performance due to the attenuation of light energy in the LGP [24,25]. Thus, despite the widespread adoption of LEDs in backlight applications, the high beam divergence of conventional LEDs limits the optical performance of backlights.

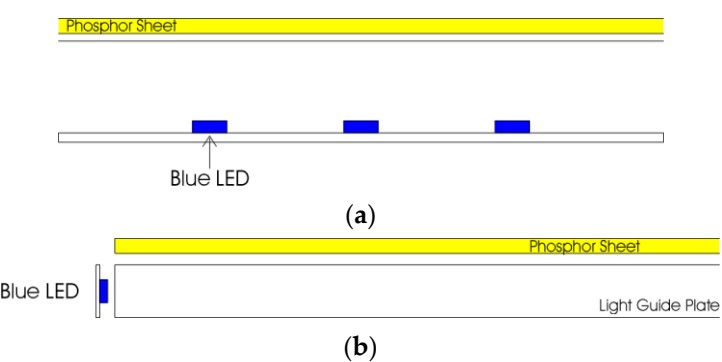

**Figure 1.** Schematic cross-sectional view of the (**a**) direct-lit backlight and (**b**) edge-lit backlight using phosphor sheet and blue LEDs.

With the recent advancements in solid-state lighting, laser diodes (LDs) have emerged as another lighting technology apart from white LEDs. Unlike LEDs, LDs primarily operate through stimulated emission when they exceed a certain threshold, resulting in light characterized by monochromaticity, coherence, high intensity, and low divergence. Integrating blue LDs with yellow phosphor enables the generation of white light with high brightness [26–28]. Inaba et al. proposed the use of laser-driven white light in an edge-lit backlight designed for mobile applications [29]. In this setup, white light produced by a blue LD and a phosphor plate propagates into the LGP to form a conventional edge-lit backlight.

In this study, we developed a remote edge-lit backlight structure by using blue LDs. Blue light emitted by blue LDs positioned at the side of the LGP passes through the LGP and spreads to the remote phosphor layer placed above the LGP, resulting in the emission of white light. Instead of dot patterns typically found at the bottom of conventional LGPs, we incorporated a diffusion reflection layer under the LGP. Our study demonstrates a practical approach to expand the output surface of the LGP by combining multiple individual LGPs, as illustrated in Figure 2a,b. Owing to the incorporation of a scattering layer between sequential LGPs, this multiple remote edge-lit backlight setup could provide uniform light output from the top surface of the combined LGPs.

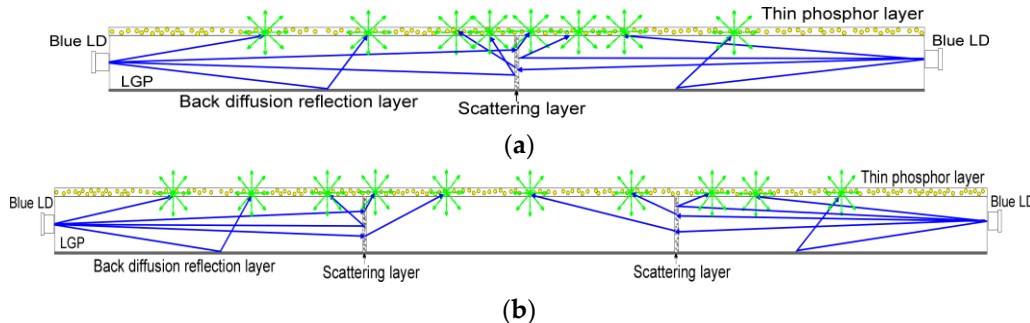

**Figure 2.** Schematic cross-sectional view of (**a**) two-LGP and (**b**) three-LGP remote edge-lit backlight structures.

## 2. Simulations of the Multiple Edge-Lit Backlight Structure

Before the demonstration of the multiple remote edge-lit backlight structure, we conducted a ray-tracing simulation by using TracePro (LAMBDA Corp., Littleton, MA, USA) to analyze the distribution of blue light in the multiple edge-lit backlight structure. The simulation setup for the multiple remote edge-lit backlight is presented in Figure 3a,b. In the absence of a remote phosphor layer on the LGP, the multiple edge-lit backlight structure comprised two or three LGPs, each equipped with a back diffusion reflection layer and eight blue LDs. A glass plate (BA270) measuring 50 mm × 50 mm × 5 mm served as the LGP for the multiple edge-lit backlight structure. The refractive index of the glass plate (BA270) was set to 1.52. However, the use of long-length sequential glass plates may reduce brightness at the interface between two glass plates or at the center portions of three glass plates. To address this problem, we introduced a scattering layer between the two glass plates. This layer leads to scattering if blue light passes through it, redirecting the light from the blue LDs toward the remote phosphor layer above the LGP, thereby ensuring uniform light output. For two- or three-LGP remote edge-lit backlight structures, we inserted a 0.5-mm-thick scattering layer between sequential LGPs. The refractive index of the scattering layer was set to 1.5. In the presence of different scattering coefficients, we used the bulk scatter properties of TracePro to simulate the blue light scattering of the scattering layer in the multiple edge-lit backlight structure. In general, Rayleigh scattering refers to the scattering of light by small particles with a size that is less than the wavelength of the light. Mie scattering describes how light scatters off a particle when the particle size is larger than the wavelength of scattered light. The Henyey–Greenstein function is commonly used to describe Mie scattering distributions due to its simple analytic form. In this study, the scattering distribution function with the Henyey–Greenstein model implemented in TracePro was used to simulate the bulk scatter properties, as given by Equation (1)

$$p(\theta) = \frac{1 - g^2}{4\pi(1 + g^2 - 2g\cos\theta)^{3/2}} \tag{1}$$

where $g$ is the anisotropy factor, which is the average cosine of the scattering $\theta$ for all the scattering events. Considering the blue light scattering of the scattering layer was isotropic, the $g$ was set to 0 in this study.

Two rows of blue LDs (OSRAM PL450B, ams-OSRAM AG, Premstaetten, Austria) operating at a peak wavelength of 450 nm were attached to the opposite edges of the multiple remote edge-lit backlight structure. The total input power of the eight blue LDs was 6.4 W (0.8 W for each blue LD). The divergence angles along the perpendicular and parallel transverse directions of the LDs were not identical. In this study, the full width at half maximum values of the PL450B LD along each direction were assumed to be 22° and 12°. Therefore, the light distribution of the PL450B can be modeled considering both divergence angles (Figure 4). In addition, the perpendicular transverse directions of all the LDs were set to be parallel to the bottom surface of the LGP (Figure 3).

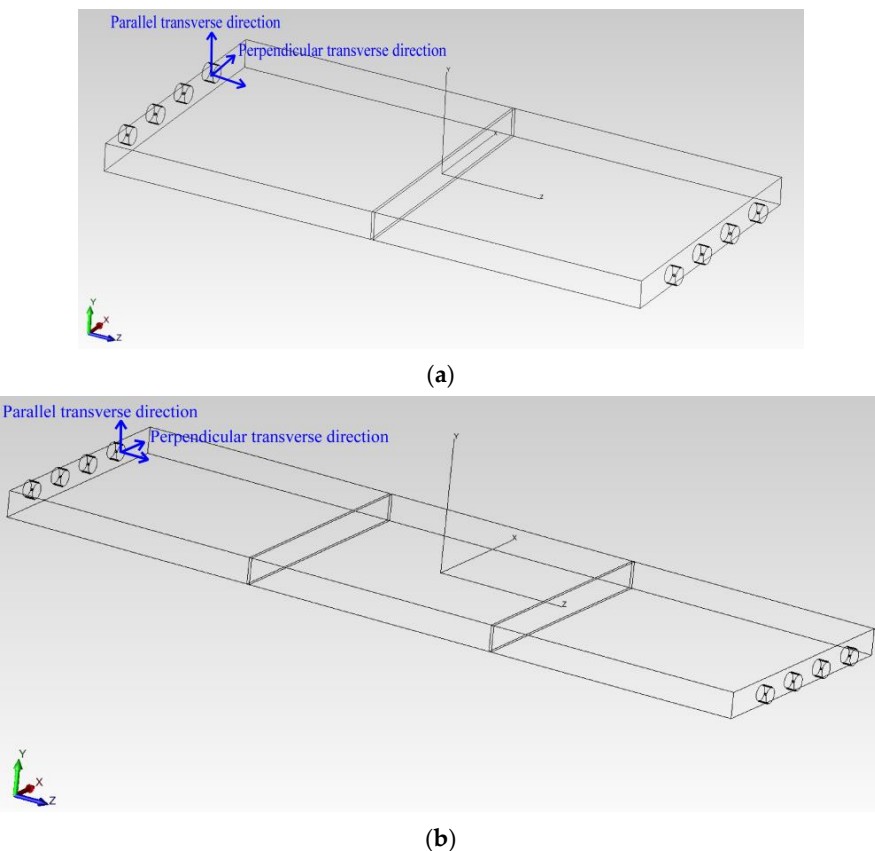

(**a**)

(**b**)

**Figure 3.** Schematic cross-sectional view of (**a**) two-LGP and (**b**) three-LGP edge-lit backlight structures.

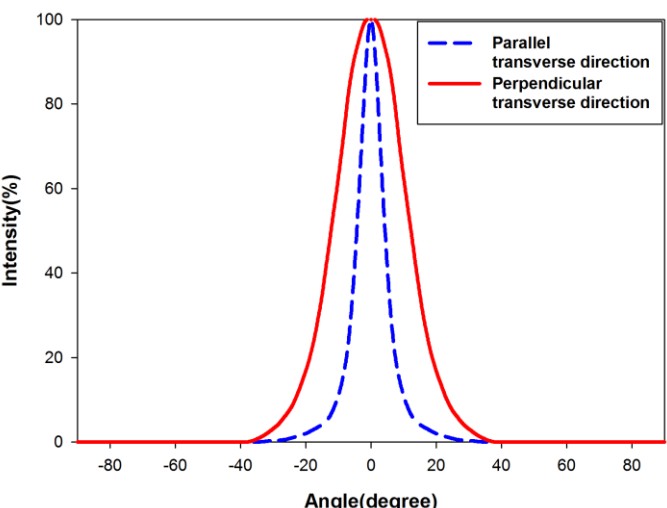

**Figure 4.** Light distribution of the blue LD used in this study.

Figure 5a displays the spatial irradiance distribution of blue light and normalized irradiance curves along the vertical and horizontal directions with a scattering coefficient of 0 (1/mm) for the scattering layer in a two-LGP edge-lit backlight structure. The two-dimensional blue light irradiance map and the normalized irradiance curves revealed that the illuminance uniformity in the central area was still not satisfactory. A fluctuation in illuminance was noted around the margin of the area. With an increase in the blue light scattering of the scattering layer (scattering coefficient: 0.1 [1/mm]), more blue light escaped from the central portion of the area, improving blue light illuminance uniformity across the entire area, as depicted in Figure 5b. However, a further increase in the scattering

of blue light within the scattering layer (scattering coefficients: 0.3, 0.5, and 0.7 [1/mm]) led to excessive blue light illuminance near the central portion of the area, reducing illuminance uniformity across the entire area, as shown in Figure 5c–e.

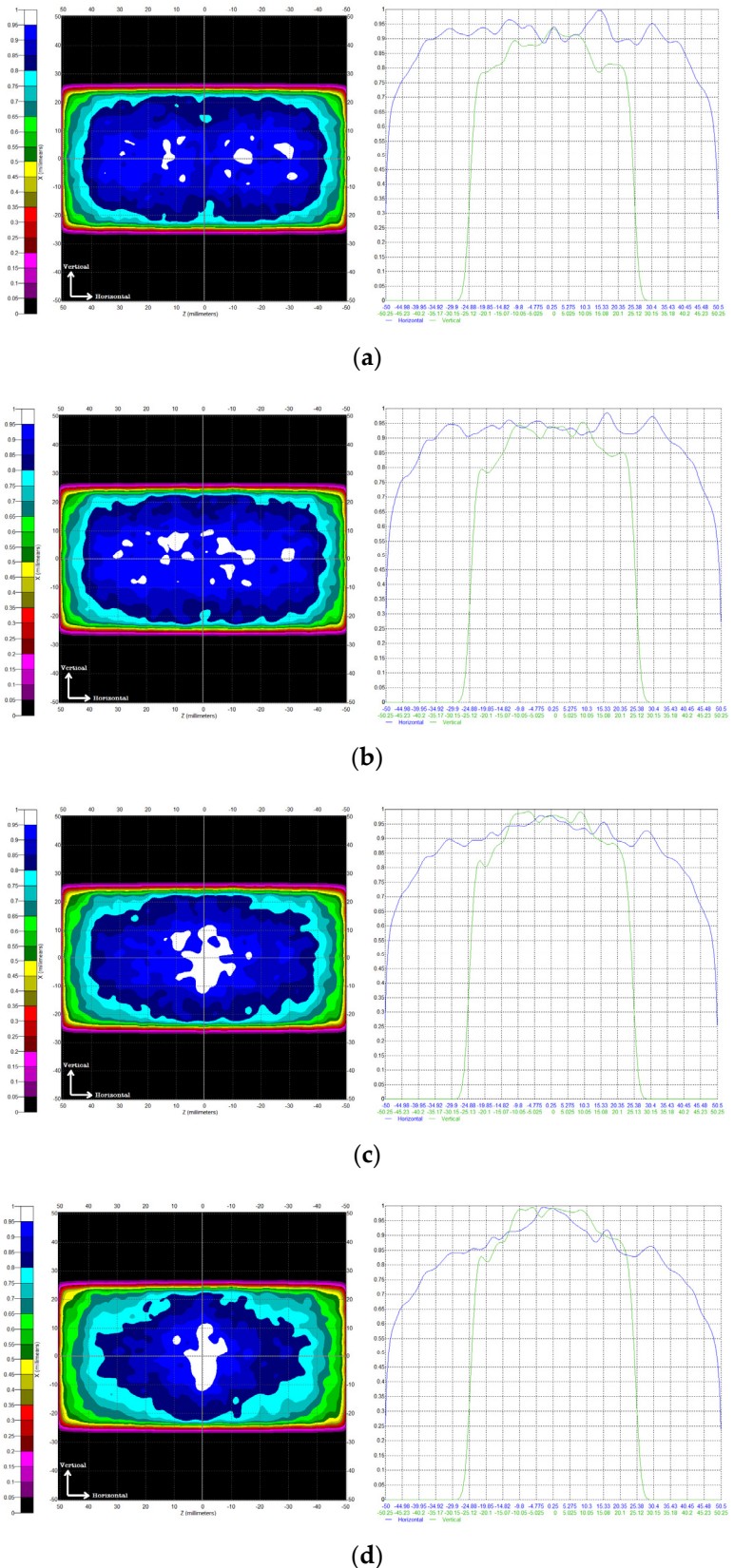

(**a**)

(**b**)

(**c**)

(**d**)

**Figure 5.** *Cont.*

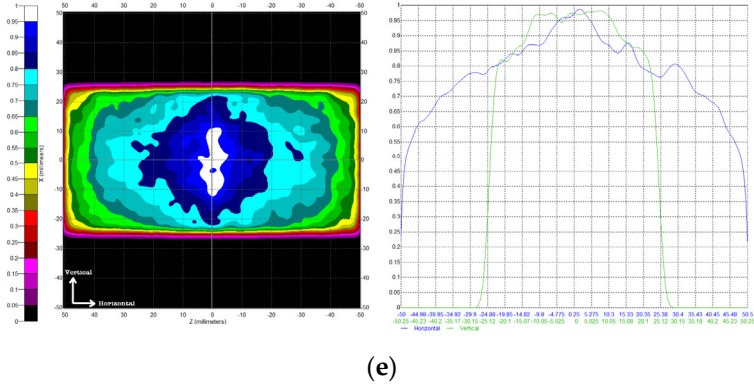

(**e**)

**Figure 5.** Simulation results of 2D blue light irradiance illuminance maps and normalized blue light irradiance distribution curves of the two-LGP edge-lit backlight structure with scattering coefficient of (**a**) 0 (1/mm), (**b**) 0.1 (1/mm), (**c**) 0.3 (1/mm), (**d**) 0.5 (1/mm), and (**e**) 0.7 (1/mm).

Figure 6a presents the spatial irradiance distribution of blue light and the normalized irradiance curves along both vertical and horizontal directions (scattering coefficient: 0 [1/mm] for scattering layers) in a three-LGP edge-lit backlight structure. The two-dimensional blue light irradiance map and the normalized irradiance curves revealed that blue light illuminance in the central area was low. This low blue light illuminance in the central region adversely affected the overall uniformity of blue light illuminance across the entire area in the three-LGP edge-lit backlight structure. As the blue light scattering of the scattering layer increased (scattering coefficient: 0.3 [1/mm]), more blue light escaped from the two scattering layers, resulting in enhanced blue light illuminance around the central area, as depicted in Figure 6b. A further increase in blue light scattering within the scattering layer, with a scattering coefficient of 0.5 (1/mm), led to even higher blue light illuminance around the central area, resulting in enhanced blue light illuminance uniformity across the entire area, as shown in Figure 6c. When the scattering coefficient of the scattering layer increased from 0.5 to 0.7 (1/mm), excessive blue light escaping from the two scattering layers led to enhanced blue light illuminance around the central area and reduced blue light illuminance uniformity across the entire area in the three-LGP edge-lit backlight structure, as depicted in Figure 6d,e. Thus, the use of a scattering layer is crucial for ensuring uniform illuminance in multiple edge-lit backlight structures.

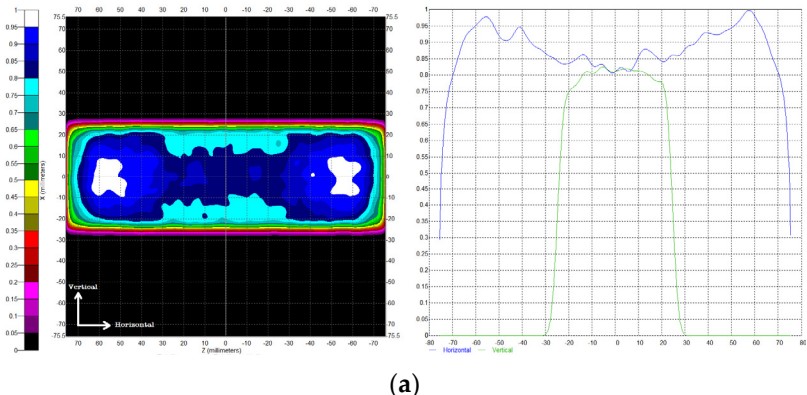

(**a**)

**Figure 6.** *Cont.*

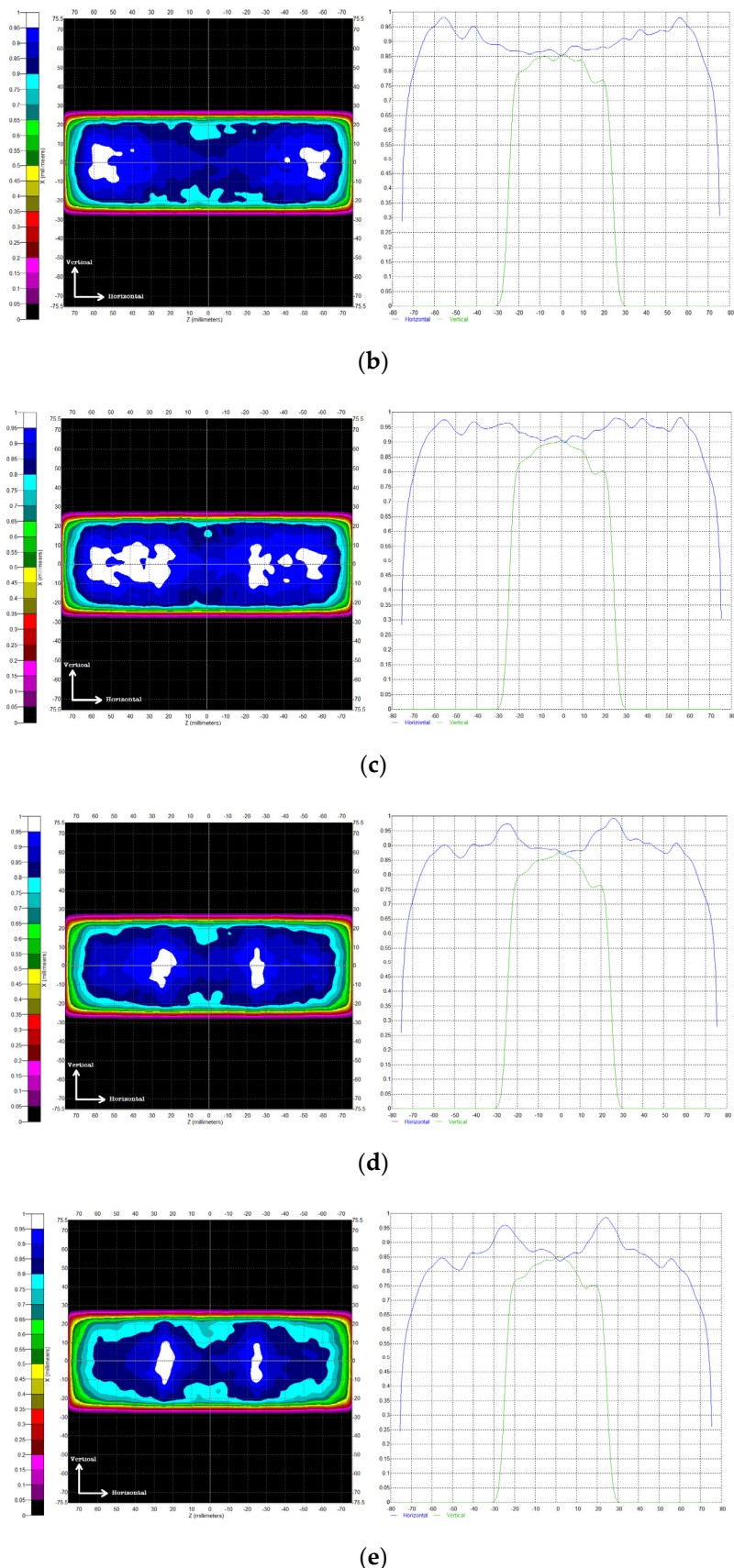

**Figure 6.** Simulation results of 2D blue light irradiance illuminance maps and normalized blue light irradiance distribution curves of the three-LGP edge-lit backlight structure with scattering coefficient of (**a**) 0 (1/mm), (**b**) 0.1 (1/mm), (**c**) 0.3 (1/mm), (**d**) 0.5 (1/mm), and (**e**) 0.7 (1/mm).

### 3. Experimental Section

To implement the multiple remote edge-lit backlight structure in this study, silicone gel (SYLGARD 184, Dow Chemical Company, Midland, MI, USA) was used to combine the sequential two or three BA270 glass plates. To induce light scattering in the adhesive layer, transparent silicon dioxide ($SiO_2$) nanopowder (Nanostructured & Amorphous Materials, Inc., Houston, TX, USA; average diameter: 1–3.5 μm) was introduced into the silicone gel SYLGARD 184 to attach the sequential two or three glass plates. Blue and white light uniformity of the multiple remote edge-lit backlight with different concentrations of $SiO_2$ nanopowder were investigated. Instead of dot patterns at the bottom of the conventional LGP, the diffusion reflection layers were applied onto the side and bottom surfaces of the LGP in the remote edge-lit backlight structure. The diffusion reflection layer on the bottom surface of the LGP directs a substantial fraction of light from blue LDs to the phosphor layer above the LGP, enabling the generation of white light. A mixture of titanium dioxide ($TiO_2$) micropowder (US Research Nanomaterials, Inc., Houston, TX, USA) and silicone gel SYLGARD 184 was spread onto the side and bottom surfaces of the LGP to form diffusion reflection layers. After the curing of silicone gel, the diffusion reflection layers were coated onto the side and at the bottom surfaces of the LGP, as shown in Figure 7a,b. To fabricate the remote edge-lit backlight structure with a phosphor layer coated on a glass plate, commercial yellow phosphor powder YAG4EL (Intematix Inc., Fremont, CA, USA) with a concentration of 3% was dispersed in SYLGARD 184, uniformly stirred with a stirrer, and poured in a rectangular aluminum mold on a glass plate. The thickness of the phosphor layer was 2 mm; it was regulated by controlling the depth of the rectangular aluminum mold. Then, the glass plate with the phosphor layer was heated in an oven at 125 °C for 20 min to cure the phosphor layer. Thus, a thin phosphor layer was produced on the glass plate, as depicted in Figure 8a,b. Two rows of blue LD (OSRAM PL450B) operating at a peak wavelength of 450 nm were attached to the opposite edges of the multiple remote edge-lit backlight structure; these diodes projected their light horizontally into the structure. The total input power of eight blue LDs was 6.4 W (0.8 W for each LD) at 120 mA. In this study, the perpendicular transverse directions of LDs were set to be parallel to the bottom surface of the LGP, as shown in Figure 3. A total of 12 and 19 points spread all over the entire two- and three- LGP remote edge-lit backlight structures were measured to evaluate the spatial luminance uniformity with a Hyperion luminance meter (Admesy, Ittervoort, The Netherlands), as shown in Figure 7a,b. The spatial luminance uniformity was evaluated as follows: Lmin/Lmax × 100%. We studied the spatial luminance uniformity of both blue light (without a remote phosphor layer) and white light (combination of blue and re-emitted yellow light) in multiple remote edge-lit backlight structures to identify the potential correlation between the blue light incident on the remote phosphor layer and the white light produced in structures with varying concentrations of $SiO_2$ nanopowder.

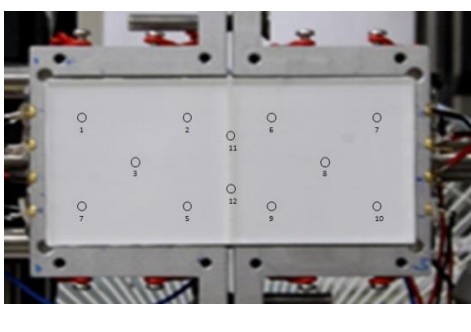 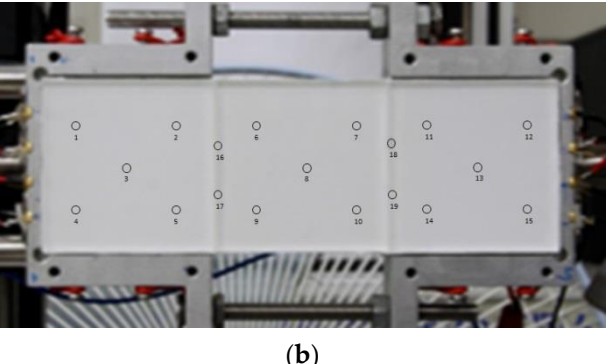

(**a**)            (**b**)

**Figure 7.** Photographs of (**a**) two- and (**b**) three-LGP edge-lit backlight structures with measurement points for uniformity evaluation.

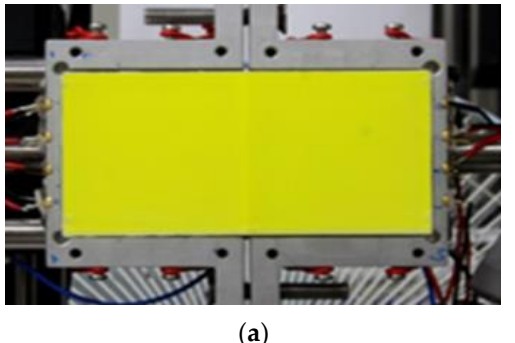 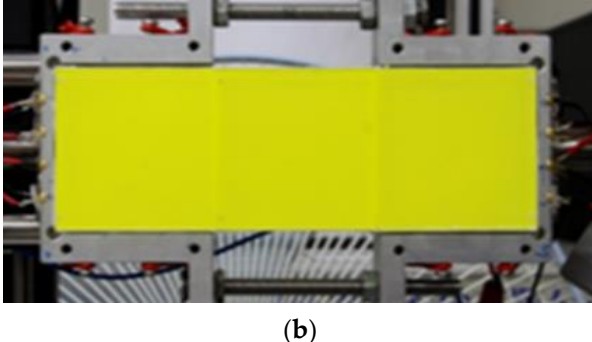

(**a**) (**b**)

**Figure 8.** Photographs of the (**a**) two- and (**b**) three-LGP remote edge-lit backlight structures.

## 4. Results and Discussion

Figure 9a,b present photographs of the blue light (without a remote phosphor layer) and white light output of a two-LGP remote edge-lit backlight structure with a $SiO_2$ nanopowder concentration of 10%. Blue light passing through the scattering layer between the two glass plates could be scattered and redirected to the top surface of the LGP, resulting in uniform light output. The concentration of $SiO_2$ nanopowder in the scattering layer is a crucial factor for the performance of a multiple remote edge-lit backlight structure. The presence of $SiO_2$ nanopowder can affect the blue light scattering of the scattering layer, potentially causing additional blue light leakage from the scattering layer and resulting in high luminance uniformity. Table 1 summarizes the measured spatial luminance uniformity for both blue and white light in a two-LGP remote edge-lit backlight structure with different concentrations of $SiO_2$ nanopowder. For a two-LGP remote edge-lit backlight structure with a $SiO_2$ nanopowder concentration of 0%, the absence of light scattering at the interface between the two glass plates resulted in low blue brightness in the central portion of the structure, leading to low spatial blue light uniformity. Blue light scattering increased with the increasing concentration of $SiO_2$ nanopowder (to 5% and 10%), leading to additional blue light leakage from the scattering layer, which improved blue light brightness in the central portion and enhanced spatial blue light uniformity. However, a further increase in the $SiO_2$ nanopowder concentration (to 15% and 20%) led to excessive blue light leakage at the interface, subsequently reducing spatial blue light uniformity. In the two-LGP remote edge-lit backlight structure, the blue light emitting from blue LDs on the side of the LGP passes through the LGP and spreads to the remote phosphor layer above the LGP to provide white light output. Therefore, the white light spatial luminance uniformity of the two-LGP remote edge-lit backlight structure should be closely related to the blue light spatial luminance uniformity of this structure. The white light spatial luminance uniformity of the two-LGP remote edge-lit backlight structure at different concentrations of $SiO_2$ nanopowder was similar to the blue light spatial luminance uniformity. The two-LGP remote edge-lit backlight structure with a $SiO_2$ nanopowder concentration of 10% achieved high blue and white light spatial luminance uniformities.

**Table 1.** The blue and white light spatial luminance uniformities of the two-LGP remote edge-lit backlight structure.

| $SiO_2$ Nanopowder Concentration (%) | Blue Light Spatial Luminance Uniformity (%) | White Light Spatial Luminance Uniformity (%) |
|---|---|---|
| 0 | 71.3 | 74.3 |
| 5 | 75.4 | 76.4 |
| 10 | 78.5 | 79.9 |
| 15 | 65.9 | 68.5 |
| 20 | 63.9 | 64.1 |

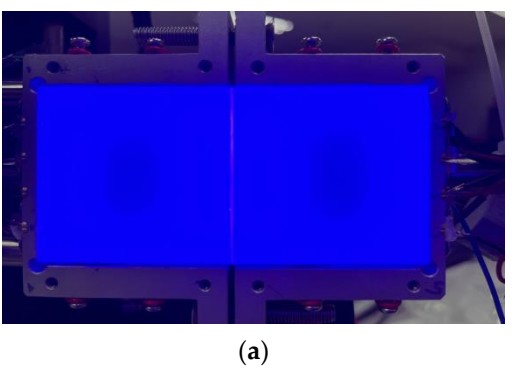 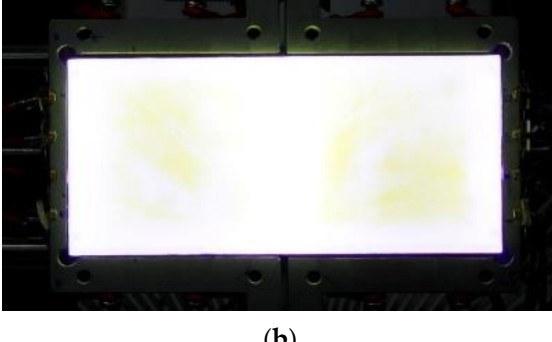

(**a**)          (**b**)

**Figure 9.** Photos of (**a**) the blue light output of the two-LGP edge-lit backlight structure and (**b**) the white light output of the two-LGP remote edge-lit backlight structure.

Figure 10a,b present photographs of the blue light (without a remote phosphor sheet) and white light output of a three-LGP remote edge-lit backlight structure with a $SiO_2$ nanopowder concentration of 10%. Blue light passing through the scattering layer between the two glass plates could be scattered and redirected from the blue LDs to the top surface of the LGP. Therefore, the proposed three-LGP remote edge-lit backlight structure can provide a uniform white light output. The measured blue and white light spatial luminance uniformities of the three-LGP remote edge-lit backlight structure with different concentrations of $SiO_2$ nanopowder are summarized in Table 2. For this structure with a $SiO_2$ nanopowder concentration of 0%, the absence of light scattering at the interface between the two glass plates resulted in low brightness near the central portion, resulting in reduced spatial luminance uniformity. Light scattering increased with the increasing concentration of $SiO_2$ nanopowder (to 5% and 10%), resulting in additional blue light leakage at the interface. The increased blue light brightness around the central portion of the three-LGP remote edge-lit backlight structure and enhanced spatial luminance uniformity. However, a further increase in the concentration of $SiO_2$ nanopowder (to 15% and 20%) led to excessive blue light leakage at the interface. For the three-LGP remote edge-lit backlight structure, the blue light emitting from blue LDs on the side of the LGP passes through the LGP and spreads to the remote phosphor layer above the LGP, generating white light output. Therefore, the white light spatial luminance uniformity of this structure may be closely related to its blue light spatial luminance uniformity. The white light spatial luminance uniformity of the three-LGP remote edge-lit backlight structure at different $SiO_2$ nanopowder concentrations was similar to its blue light spatial luminance uniformity. The three-LGP remote edge-lit backlight structure with an $SiO_2$ nanopowder concentration of 10% achieved high blue and white light spatial luminance uniformities.

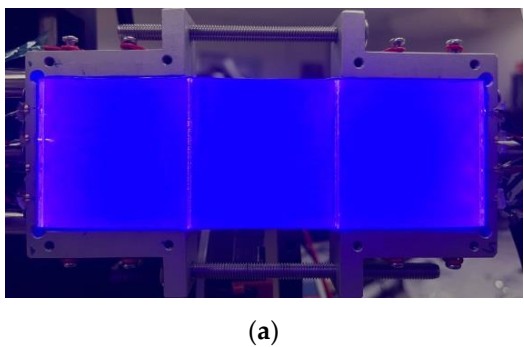 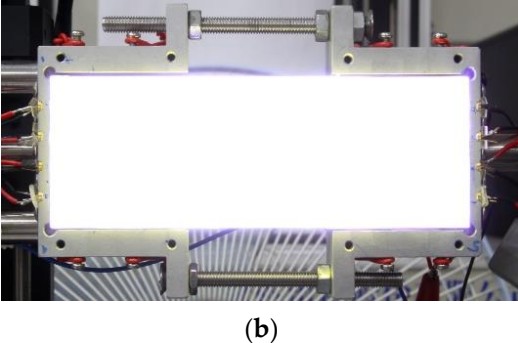

(**a**)          (**b**)

**Figure 10.** Photos of (**a**) the blue light output of the three-LGP edge-lit backlight structure and (**b**) the white light output of the three-LGP remote edge-lit backlight structure.

**Table 2.** The blue and white light spatial luminance uniformities of the three-LGP remote edge-lit backlight structure.

| SiO$_2$ Nanopowder Concentration (%) | Blue Light Spatial Luminance Uniformity (%) | White Light Spatial Luminance Uniformity (%) |
|:---:|:---:|:---:|
| 0 | 47.0 | 51.8 |
| 5 | 62.5 | 67.0 |
| 10 | 78.4 | 81.6 |
| 15 | 63.0 | 64.7 |
| 20 | 58.6 | 58.7 |

By using a blue LED array and yellow remote phosphor film, Ito et al. proposed a planar white light for direct-lit backlight applications [20,21]. With an optimized configuration of the lighting module in the planar lighting system using a blue LED array to excite yellow remote phosphor film, Huang and Tien et al. also demonstrated a planar lighting system with high uniformity [19,22]. With the planar lighting system integrated with a 7″ TFT-LCD panel, a light-emitting uniformity of 82% can be achieved. With a similar structure, Chen et al. proposed a uniform white direct-lit backlight using mini-chip-scale packaged light-emitting diode (mini-CSPLED) and quantum dot (QD) films [18]. The backlight with a size of 18 mm × 18 mm contained a 3 × 3 mini-LED array, diffusion plate, QD films, and two prism films. By using a mini-CSPLED with an emission angle of 180° and a 150 μm thick QD film, a brightness uniformity of approximately 86% for planar white light can be achieved. In this study, the prototypes of two- and three-LGP edge-lit backlight structures demonstrated a white light illuminance uniformity of 79.9% and 81.6%, respectively. Although the white light illuminance uniformities of two- and three-LGP edge-lit backlight structures are a little lower than the planar lighting system and mini-CSPLED backlight unit, the proposed architecture presents a viable solution for achieving good uniformity in planar lighting systems using blue LDs.

## 5. Conclusions

This paper presents a novel remote edge-lit backlight structure involving blue LDs. Blue light emitting from blue LDs on the side of the LGP passes through the LGP and spreads to the remote phosphor layer above the LGP, generating white light output. The incorporation of a scattering layer between sequential LGPs offers a feasible means to expand the output surface of the LGP by combining multiple individual LGPs. Through the blue light simulation of multiple edge-lit backlight structures, both two- and three-LGP edge-lit backlight structures could achieve acceptable blue light illuminance uniformity across the entire area. The prototypes of two- and three-LGP edge-lit backlight structures demonstrated a blue light illuminance uniformity of 78.5% and 78.4%, and a white light illuminance uniformity of 79.9% and 81.6%. Although the uniformity of the prepared prototype is still not good enough at this moment, the proposed architecture presents a viable solution for achieving planar lighting systems with blue LDs, particularly suitable for linear lighting or slender backlighting applications instead of display stand applications.

**Author Contributions:** Conceptualization, S.-Y.T. and Y.-K.C.; methodology, B.-M.C. and S.-P.Y.; resources and supervision, B.-M.C.; formal analysis, visualization, writing—original draft, writing—review and editing, S.-P.Y.; investigation, software and data curation, T.A.P.; Validation, S.-Y.T. and Y.-K.C. All authors have read and agreed to the published version of the manuscript.

**Funding:** The authors would like to thank the National Science and Technology Council of Taiwan for their financial support under grant number NSTC 112-2637-E-159-002.

**Institutional Review Board Statement:** Not applicable.

**Informed Consent Statement:** Not applicable.

**Data Availability Statement:** Data are contained within the article.

**Conflicts of Interest:** The authors declare no conflicts of interest.

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
