# Peer review of "The Development of a Remote Edge-Lit Backlight Structure with Blue Laser Diodes"

_photonics, doi:10.3390/photonics11010078_

Round 1

Reviewer 1 Report

Comments and Suggestions for Authors

This manuscript introduced a novel design of a remote edge-lit backlight structure featuring blue laser diodes (LDs). These LDs were integrated into a remote yellow phosphor layer on a light guide plate (LGP). Blue light emitted by the LDs passes through the LGP and spreads to the remote phosphor layer, generating white light output. Owing to the incorporation of a scattering layer between sequential LGPs, the remote edge-lit backlight structure facilitates the expansion of the output surface of the LGP by combining multiple individual LGPs. Two- and three-LGP remote edge-lit backlight structures demonstrated high white illuminance uniformity. Although this new structure is somewhat innovative, I think it is still difficult to realize the actual application. I cannot recommend publication of this manuscript in a present form. This manuscript should be accepted after revisions commented on below.

1. The authors combined multiple individual LGPs into a new remote edge-lit backlight structure by using two rows of blue LD. This new structure in the actual production application needs to redesign the components instead of using existing components, which undoubtedly increases the production cost. This question should be considered.

2. In this paper, the glass plate (BA270) measuring 50 mm × 50 mm × 5 mm served as the LGP for the multiple edge-lit backlight structure. As far as I know, the thickness of the ordinary LGP is 2 mm or even thinner, and the thickness of 5 mm used by the author makes the backlight structure thicker. How does the authors consider this issue?

3. The paper does not account for the refractive index of the glass plate and suggests an addition. The authors have mentioned the scattering coefficients of the scattering layer, how to determine the scattering coefficient of the scattering layer?

4. The authors said that the two-dimensional blue light irradiance map and the normalized irradiance curves in Figure 5 and 6 revealed that the illuminance uniformity in the central area was satisfactory. However, the two-dimensional blue light irradiance maps in Figures 5 and 6 do not look uniform, and there are obvious gaps at the splices of the 2 and 3 LGPs edge-lit backlight structure in Figures 9 and 10, which also look uneven. Please explain this phenomenon.

5. The maximum uniform value of white light illuminance of the prototype is 81.6%, however, the commercial backlight illuminance uniformity is generally more than 85%, the uniformity of the prepared prototype is still not good enough. The authors should make the results more convince.

6. English language of the whole paper requires careful grammar cheking, such as, page 1, line 37, “LCDs do require a backlight” should be changed to "LCDs require a backlight".

7. The authors' names in Reference 10, 22 and 27 should be properly indicated, and the title format of the references should be consistent, with some having capitalized initial letters and some not.

Comments on the Quality of English Language

Minor editing of English language required

Reviewer 2 Report

Comments and Suggestions for Authors

The reviewed manuscript is devoted to the development of a remote side LED backlight design.

The manuscript leaves the impression of a very superficial discussion of the results obtained.

Questions and comments on the text of the Manuscript.

1. Citations [12-17] and [18-23] should have more detailed descriptions of 1-3 references, not 6.

2. Figure 2: with a small divergence of Blue LED radiation, little light will fall on the Back diffusion reflection layer areas close to them. The radiation of the Thin phosphor layer will be uneven, as can be seen from further results. How critical is it?

3. Poor quality of graphs in figures 5 and 6.

4. What are the chromaticity coordinates of the received white light in XYZ or RGB systems?

5. What is the cost and manufacturability of using the illumination system proposed by the authors?

6. Section 4 Results and discussions should be strengthened. It is necessary to make a comparison with other types of illumination.

Reviewer 3 Report

Comments and Suggestions for Authors

The paper reports on developing an edge-lit backlight structure featuring blue laser diodes (LDs) for display applications. The manuscript is well-written and organized with a focus on contemporary technological display applications. It offers valuable technical details and guidance for improving the performance of a remote edge-lit backlight display structure with LDs. Scientific discussion is limited and should be improved by offering further insight into the physics of blue light scattering processes induced by scattering interfaces between LGPs.

Selected issues to be addressed by the authors:

Please define a scattering coefficient.

Does the physical interface quality between LGP/scattering layer/LGP affect the blue light coupling between media having different refractive indexes?

Fig.2 shows blue light scattering by the scattering layer resulting in light propagation (forward). Is there any blue light backscattering by the scattering layer? The light rays indicating such scattering are not shown in Fig.2. Why?

Please comment on using smaller SiO2 particles than 1um diam. for the scattering layer.

Some discussion focused on Mia vs Rayleigh scattering would be beneficial here.

Please comment on the origin of reduced illuminance around the margin of the area. How to eliminate this drawback and improve the illuminance homogeneity across the display?

In conclusion, we recommend minor revisions before accepting the manuscript.   

Comments on the Quality of English Language

Minor editing of English language required.

Round 2

Reviewer 1 Report

Comments and Suggestions for Authors

The authors revised to the questions and the comments. While most of the revisions are adequately addressed, the authors' names in Reference 10, 22 and 27 are still not properly indicated, and the title format of the references are still not consistent. The authors' names in Reference 10, 22 and 27 should be no cross. The references 3,4,7,8,12,14,16,18,19,21,23,29 are headline style capitalization, while others are sentence style capitalization

Comments on the Quality of English Language

Minor editing of English language required